# Infection Control Experiences and Educational Needs of Geriatric Care Workers in Long-Term Care Facilities: A Pilot Study

**DOI:** 10.3390/healthcare12030301

**Published:** 2024-01-24

**Authors:** Min Sun Song, Sun Young Jeong, Soohyun Park

**Affiliations:** 1College of Nursing, Konyang University, 158, Gwanjeodong-ro, Seogu, Daejeon 35365, Republic of Korea; mssong@konyang.ac.kr (M.S.S.); jsy7304@konyang.ac.kr (S.Y.J.); 2Department of Nursing, Eulji University, Seongnam, 553 Sanseong-daero, Sujeong-gu, Seongnam-si 13135, Republic of Korea

**Keywords:** long-term care, focus group, infection control, caregivers, education

## Abstract

Background: In the post-COVID-19 condition, infection control education is important for geriatric care workers who care for the elderly and are vulnerable to emerging infectious diseases. This study was conducted to enhance the insight into the experiences of geriatric care workers in managing novel infectious diseases (COVID-19) and to identify the newly required educational requirements necessary to effectively implement infectious disease control. Methods: This is a qualitative and pilot study using focus group interviews. Data from 10 participants were collected using a focus group interview. The data were analyzed using Qualitative content analysis. Results: The findings showed that geriatric healthcare workers experienced difficulties following infection control protocols and emotional distress related to visitor restrictions and had an increased workload. The participants requested further education regarding general knowledge of infectious diseases to decrease their fears of infection and reported that visual and practical teaching methods were preferable. Conclusions: Further attention is needed regarding the education of infection control to strengthen infection prevention in long-term care facilities vulnerable to the spread of emerging infectious diseases.

## 1. Introduction

The COVID-19 pandemic, which lasted for more than three years, was ended by the World Health Organization on 5 May 2023. Nevertheless, healthcare providers must be prepared to care for emerging infectious diseases like COVID-19. During the COVID-19 pandemic, it was found that older adults were vulnerable to emerging respiratory infectious diseases. The mortality rate in the ages of 80 years and above is very high (58.89%), with a fatality rate of 2.68% [1]. In particular, older adults in long-term care (LTC) facilities are at a high risk of infectious disease outbreaks, since most residents are medically fragile [2,3,4]. In the Republic of Korea, more than 65% of adults in LTC facilities were over age 80, and in the LTC facilities with confirmed cases of COVID-19, 65% reported clusters of confirmed cases [5].

According to policies related to COVID-19 infections in LTC facilities, case clusters were related to facilities having poor infection control systems and a lack of official guidelines or education programs addressing infectious disease prevention [5,6]. Moreover, older adults in LTC facilities are often at a heightened risk of infection due to difficulties in providing additional quarantine spaces, inadequate preventive measures, and insufficient resources to recover from COVID-19 [7]. Therefore, strategies to protect older adults living in LTC facilities from emerging infectious diseases such as COVID-19 infection are crucial, and infection control programs specializing in LTC facilities should be considered [6,8].

Numerous nations have established and implemented infection prevention and control initiatives designed for personnel in LTC facilities. However, direct health care providers in LTC facilities were more likely to perceive infection prevention and control as low priority rather than their responsibility. Furthermore, they have a limited education about infection control and prevention [9]. In the Republic of Korea, geriatric care workers, certified by the Ministry of Health and Welfare, are responsible for providing care to infected patients. These workers have not received a formal education from a professional medical institute but completed a short-term program lasting only 240 h that includes the theory and practice related to the care of frail older adults [10]. According to the training guidelines for geriatric care workers in LTC facilities, they are required to participate in continuing education and training once a year. However, these training programs do not include standardized educational guidelines for infection prevention and control, which do not exist [11].

Previous research has reported that having staff who are either underqualified or have inadequate infection control training represents a leading cause of increased COVID-19 spread [6]. Less than 3% of LTC facilities provided opportunities for healthcare workers to attend infection control courses compared to such opportunities that were available in over 90% of acute care hospitals [12]. No study has examined the educational needs of geriatric care workers in LTC facilities, although education was the most straightforward and effective measure to control infection [13,14]. Given the significant impact of COVID-19 and other healthcare associated infections in LTC, it is essential to understand the difficulties experienced by geriatric care workers related to infection control to better protect residents and geriatric care workers in LTC facilities. Furthermore, this study can be the baseline for developing infection control programs.

This study aimed to conduct focus group interviews to understand the geriatric care workers’ experiences while working to control emerging infectious diseases (i.e., COVID-19) and to identify their educational needs for developing systematic education programs regarding the newly required infectious disease controls.

## 2. Materials and Methods

### 2.1. Design

This qualitative and pilot study used focus group interviews to assess the experience of geriatric care workers related to infection control and to identify their educational needs related to the emerging infectious diseases in LTC facilities.

### 2.2. Participants

The focus group participants were recruited using the snowball sampling method. The managers who had participated in an educational program for LTC facility managers suggested eligible participants. A researcher invited 10 geriatric care workers to participate in this study and answer related questions. All ten geriatric care workers in LTC facilities were recruited. Participants were from the southwest regions of the Republic of Korea and were divided into two groups based on their job roles and duties: (1) five who work for managers; and (2) five who provided direct patient care. Participants were eligible to participate if they had previous experience in nursing care and/or the management of older adults with emerging infectious diseases, including COVID-19, as a geriatric care worker in LTC facilities.

### 2.3. Ethical Considerations

This study was approved by the Institutional Review Board (IRB No: KYU 2021-08-009-001) of the Konyang University in Korea. The purpose and method of the study were explained to participants before the focus group interview was conducted. Participants provided their written informed consent after they were informed that their participation was entirely voluntary, that they could refuse or discontinue their participation at any time, and that there was no disadvantage to refusing to participate. Researchers explained the confidentiality of the participants’ information, and all participants approved the video recording of their interviews before the focus group interviews were conducted on the ZOOM online platform. Participants received a $15 reward in USD after completing the interview. The reward was intended as an expression of gratitude for the time devoted to the interview and was not associated with any benefits for the study. Identification of participants was coded using pseudonyms, giving an initial letter per participant in alphabetical order. After the focus group interview, the data, including audio recordings and transcripts, were encrypted and secured using passwords and secure storage devices to prevent unauthorized access for three years, and after that, the data will be deleted and destroyed.

### 2.4. Data Collection

Focus group interviews were conducted twice, on the 15th and 16th of September 2021. The interviews lasted for two hours. The primary investigator developed the content for the focus group interview, including a series of semi-structured interview questions and prompts (Table 1). Five experts, including three LTC facility managers and two geriatric nurse practitioners specializing in infection control, reviewed the content of the questions.

The interview was facilitated by two researchers who explained the purpose and process of the study and guided the direction of the interview for the participants. The interviews lasted for up to two hours until the data reached saturation. The focus group interview was conducted via the ZOOM (Zoom Video Communications, Inc., San Jose, CA, USA) online platform for safety reasons related to the COVID-19 pandemic. The interview sessions were recorded using the ZOOM recording system.

### 2.5. Data Analysis

To analyze and organize the qualitative data, we used qualitative content analysis, as suggested by White and Marsh [13], which has been widely used in social sciences. The analysis process was followed by the following main steps: (1) multiple sequential readings of the answers: the interview recordings were transcribed in a verbatim manner and thoroughly reviewed. (2) Extraction of the meaning codes: the statements underwent coding, categorization, and abstraction processes. (3) Categorization and abstraction: similar codes were clustered into the same sub-categories. (4) Analysis and interpretation: the researchers collaborated to review one another’s opinions and agreed on the final categories derived.

The coded sentences were divided into categories and labeled, then organized into five key themes and seven sub-themes. The final themes and sub-themes were revised until the two researchers reached an agreement by comparing the meaningful statements and content and analyzing the themes and sub-themes.

The reliability and validity by Kidd and Parshall [15] were used for this study. In order to secure reliability, the criteria of stability, equivalence, and internal consistency were used. In this study, stability was ensured by interviewing two groups (managers and geriatric care workers) once each. Equivalence was secured by repeatedly reading and comparing the collected data until two researchers agreed on the meaningful statements derived and the coded category analysis results. Internal consistency was secured by comparing and confirming the interview contents between the focus groups. The validity of the data analyzed from the interviews was verified if the data analyzed by one of the interview participants well reflected the experience of infection control and the educational needs of geriatric care workers in LTC facilities.

## 3. Results

### 3.1. Participant Characteristics

The focus group participants included five geriatric care workers who were managerial staff and five who were direct patient care providers in LTC facilities. Participants had a mean age of 61.6 years (*SD* = 5.32, range 60–67) and a mean length of experience working in an LTC facility of 10.3 years (*SD* = 5.87). The participants’ education levels included five who had completed graduate school (managerial staff), one university graduate, and four high school graduates. The LTC facilities where the participants worked had 68 to 109 residents. None of the participants had received formal education related to infection prevention and control.

### 3.2. Focus Group Themes

Analysis of the focus group interviews revealed three main themes: (1) “geriatric care workers’ infection control job duties in LTC facilities”, (2) “geriatric care workers’ experience with prevention and control of emerging infectious diseases”, and (3) “geriatric care workers’ educational needs related to prevention and control for emerging infectious diseases” (Figure 1).

**Theme** **1.**
*Geriatric care workers’ infection control job duties in LTC facilities.*


After analyzing the transcripts of the geriatric care workers’ interviews, 22 meaningful codes for Theme 1 were generated, and the codes were grouped into five sub-themes based on code similarity. The duties performed by geriatric care workers in LTC facilities that were related to infection control included: (1) checking whether personal protective equipment (PPE) was worn correctly, (2) checking and reporting on the residents’ health status, (3) environmental cleaning, disinfecting, and ventilation, (4) monitoring colleague’s hand hygiene, and (5) completing administrative documentation and paperwork.

(1)Checking whether PPE was worn correctly.

Geriatric care workers expressed that while they attempted to place masks over older residents’ mouths and noses in the LTC facilities, many of the older adults had difficulties with the continued use of masks. Furthermore, older adult residents complained of breathing-related discomfort while wearing their masks. Therefore, geriatric care workers were required to check on residents continually regarding their mask use.

“*There are only a few people who wear masks, and most residents cannot wear them because of physical discomfort, such as difficulty breathing. In addition, they frequently took off their masks even when they had visitors*.”(Participant C)

“*Very few older residents wore masks. Some of the older adults put their masks in their diapers or wore the masks on their heads because of dementia. Only bedridden patients could wear them all the time*.”(Participant D)

(2)Checking and reporting on the residents’ health status.

Geriatric care workers in LTC facilities needed to check the health status of older residents daily, including residents’ body temperatures, to identify COVID-19 symptoms, and were required to report these results.

“*Whenever I entered an older resident’s room, I checked their health status by regularly examining their bodies to see if they had a fever. I also checked to see if residents were experiencing discomfort after eating food, indigestion, or coughing. I checked and reported the health status to the nurse immediately*.”(Participant J)

“*Fever is the most common symptom in all the reactions to COVID-19. Therefore, I’m paying a lot of attention to checking the health status of the older residents. I concentrated on checking for fever, the condition of the older residents, and whether they have other symptoms*.”(Participant I)

(3)Environmental cleaning, disinfecting, and ventilation.

Infection control requires geriatric care workers to clean and disinfect the environments, such as the residents’ beds and floors, by wiping down the surfaces with disinfectants and ventilating the room while exchanging the diapers of older residents.

“*Disinfecting products with more than 83% Ethanol were delivered, and they were used as a disinfectant. Diluted chlorine bleach was used for laundry…and the toilet floor was cleaned with diluted bleach*.”(Participant B)

“*We’ve set up a time for ventilation. After breakfast, lunch, dinner, and changing diapers, the windows were opened for ventilation*.”(Participant A)

(4)Monitoring hand hygiene.

Geriatric care workers frequently perform hand hygiene while working and monitor each other’s hand hygiene.

“*Geriatric care workers often forget to disinfect their hands while working, so they monitor hand hygiene each other*.”(Participant B)

(5)Completing administrative documentation and paperwork.

During the COVID-19 pandemic, administrative paperwork, such as disinfection and hand hygiene reports, increased. They noted that it was difficult to work and complete paperwork at the same time.

“*Now, there is a lot of work for me related to COVID-19 documents and reports… Although geriatric care workers deal with documentation work, it is hard to complete the documents and take care of the older residents simultaneously*.”(Participant G)

“*Compared to before the COVID-19 pandemic, we have more work to do now. And it is more difficult to care for the older residents who have problematic behaviors because a lot of paperwork was added. We have to document every day all the kinds of used disinfected products. We just hope to report this to the main office, but we have to submit documents for all the related parts separately, such as to the welfare service, local government, and community health centers*.”(Participant D)

**Theme** **2.**
*Geriatric care workers’ experience with the prevention and control of emerging infectious diseases.*


We identified 31 meaningful expressions in the interviews related to Theme 2, although geriatric care workers’ experiences of managing emerging infectious diseases varied across the participants. The 31 statements were analyzed into eight sub-themes: (1) difficulty with wearing PPE, (2) difficulty with following an infection control manual, (3) families suffering from pandemic-related visitor restrictions, (4) older residents’ psychological problems related to visitor restrictions, (5) lack of communication with public health care centers, (6) difficulty with performing hand hygiene according to the manual, (7) increased workload, and (8) difficulty with transferring residents to hospitals.

(1)Difficulty with wearing PPE.

Geriatric care workers stated that they had difficulties working while wearing PPE in clinical practice during the COVID-19 outbreak. Wearing KF94 respiratory protective masks can lead to breathing and communication difficulties among older residents.

“*I always wear a mask and wash my hands. It was okay to wash my hands, but it was not easy to wear a mask in hot weather*.”(Participant J)

“*There were difficulties in communicating with older residents because I was always wearing a mask while working. When the residents’ pronunciations were unclear, we observed their mouths. However, wearing a mask interrupted it, so we could not read the lips*.”(Participant H)

(2)Difficulty with following an infection control manual.

Geriatric care workers indicated that the existing infection control manual was not feasible in real LTC facilities. Participants shared that it would be beneficial if a manual that could be used practically was developed.

“*The manual didn’t fit with the real-world situation during COVID-19. It says that the older residents should maintain a distance of 2 m to prevent infection and should be restricted from moving between levels and floors on site. If you keep a distance of 2 m, you can’t give the resident assistance during meal times and daily activities. The manual is so different from the real situation in long-term care facilities. When a COVID-19 patient was confirmed, the health care center told me not to go into the facility… We couldn’t stay without going in there. As a result, most residents were infected with the coronavirus*.”(Participant D)

(3)Families suffering from pandemic-related visitor restrictions.

During the COVID-19 outbreak, social interactions were limited to reduce the spread of the virus in LTC facilities. With severe restrictions on visitors to LTC facilities, some family members would be stubborn and demand to have face-to-face visits or discharge older residents. In addition, visitation limitations had negative consequences, such as distrusting geriatric care workers because they visit their relatives.

“*Some family members ask me for face-to-face visits whenever they come. However, since face-to-face visits were not allowed, I refused it. If they stubbornly request a face-to-face visit, I can’t handle it. I don’t want to violate the quarantine laws, too. The older resident was discharged by his family member because video call communication was only allowed*.”(Participant A)

“*If bruises were found on an older resident’s body during the period of visitor restriction, one family member threatened geriatric care workers with human rights issues*.”(Participant D)

(4)Older residents’ psychological problems related to visitor restrictions.

Many older residents were anxious and annoyed during the period of severe visitor restrictions in LTC facilities. Their depression worsened, and they occasionally refused to eat.

“*The older residents were always anxious and very annoyed when they couldn’t see their family members. Some older residents couldn’t recognize the COVID-19 outbreak, so they asked why family members did not visit them. At that time, we felt very sorry for them*.”(Participant F)

(5)Lack of communication with public health care centers.

Geriatric care workers mentioned a lack of communication with the public health care centers during the COVID-19 pandemic. They did not know how or where to transfer patients with positive COVID test results or how to care for infected patients.

“*After receiving a report about a positive COVID-19 test, the public health care center did not answer my phone calls even though the infected patient was in bad condition… The real situation was different from what the public health care center reported on the progress of COVID-19*.”(Participant D)

(6)Difficulty with performing hand hygiene according to the manual.

Geriatric care workers mentioned that they have difficulties following the manual’s instructions regarding hand hygiene because they were required to share the bathroom sink used for hand washing with many other co-workers.

“*Since the bathroom is shared, hand washing is not done frequently according to the instructions in the manual. I was supposed to disinfect my hands and frequently replace disposable gloves, but I couldn’t*.”(Participant F)

(7)Increased workload.

The workload of geriatric care workers increased during the COVID-19 pandemic. The amount of mandatory documentation increased, including a periodic report of contact tracing and daily disinfection documentation. There was also additional work related to sending online messages, such as video messages or photos, to residents’ family members.

“*All facilities require additional disinfection after COVID-19. I spray alcohol solution or diluted chlorine bleach to disinfect the bathroom every day. In addition, disinfection from an outside disinfection company was often done. Anyway, not just my facility, but most facilities have a lot of additional work to do*.”(Participant F)

“*We helped the older residents call their families, took pictures of them completing activities, and sent them to their family members. We showed the older residents pictures of their family members. There was a lot of work to do*.”(Participant I)

“*We recorded contact tracing of staff members in facilities…when we follow all the infection control protocols, I think that it’s more than we can handle with our human resources*”.(Participant C)

(8)Difficulty with transferring residents to hospitals.

Due to the COVID-19 pandemic, transferring older residents with respiratory symptoms to the hospital was difficult. Emergency departments would refuse to admit these patients.

“*The SPO_2_ dropped below the 50s, so I sent the resident to the emergency room, but every hospital didn’t get the patient because he had respiratory symptoms. So I begged several university hospitals, and the patient was finally admitted to one of the university hospitals right before he expired*.”(Participant D)

“*If you want to send a patient to the emergency room, you must call and then go to the emergency room. Also, if a patient took a PCR test and the results were not out yet, the hospital would return the patient to the facility. At that time, we were not sure whether we could let the patient stay in the facility or not*.”(Participant G)

**Theme** **3.**
*Geriatric care workers’ educational needs related to prevention and control of emerging infectious diseases.*


Participants mentioned 29 meaningful expressions related to Theme 3, and after classifying the content, three sub-themes were identified: (1) education providing a general overview of infectious diseases, (2) education to decrease fear of infection, and (3) teaching method preferences.

Geriatric care workers in LTC facilities wanted education about emerging infectious diseases, including the type of pathogens and infection sites. They preferred teaching that employed visual and practical methods such as video lessons, periodic teaching for repetition, and demonstration practice.

(1)Education providing a general overview of infectious diseases.

Geriatric care workers wanted a general knowledge of infections, such as concepts, transmission routes, symptoms, factors influencing infectious diseases, different types of pathogens, and differences in management according to the type of infectious disease (e.g., skin infection, urinary tract infection, and respiratory infection).

“*Although it’s important not to be infected with the coronavirus, I think that it’s very important to recognize my behavior impacted my workplace and my family. I think that we should know how a virus can infect us and what happens when we don’t follow the quarantine*.”(Participant B)

“*I hope to know how the virus is delivered using an animated cartoon*.”(Participant D)

“*It would be great to know what happens when geriatric care workers do not wash their hands for disinfection when changing the diapers of older residents*.”(Participant A)

“*There are many different types of germs, such as VRE (vancomycin-resistant enterococci) and CRE (carbapenem-resistant Enterobacteriaceae). So, I hope that education could be provided depending on the type of germ*.”(Participant E)

“*It is also important to manage urinary tract infections for older residents. We need infection education for each situation*.”(Participant D)

“*I heard that there are a lot of skin infections. Rashes or skin redness often appear in older residents… I’m curious about the causes of many skin problems*.”(Participant J)

(2)Education to decrease fear of infection.

Geriatric care workers requested education to eliminate their vague fears of infection.

“*My colleagues and I have vague fears about germs. I am not sure that patients’ germs are always transmitted to our bodies when we are close to patients. I hope that education would include eliminating fear. Since geriatric care workers are working to care for older residents, they don’t want to avoid infected patients unconditionally, but they might want to know how to manage it*.”(Participant E)

(3)Teaching method preferences.

Geriatric care workers preferred visual and practical teaching methods, including video lessons, periodically repeated instruction, and practice with demonstrations based on hands-on experience rather than traditional one-way lectures.

“*I think it’s good to show educational video clips that are made in the LTC facilities rather than one-way lectures by an instructor*.”(Participant J)

“*In fact, since face-to-face education is difficult because of COVID-19, I would like to say that we need education using online platforms like ZOOM. Also, we need repetition in our instruction, not just one-time education, and education that can be applied in a real-world setting*.”(Participant G)

“*There is a difference between people who participate in infection control practice and those who receive only education online*.”(Participant H)

“*I think that it will be very effective if educational animation is produced that includes the way the infectious disease spread by one careless person*.”(Participant A)

## 4. Discussion

The current pilot study was conducted using focus group interviews with a sample of geriatric care workers in LTC facilities to explore their experience and educational needs regarding infection control to develop an infection control education program.

Geriatric care workers participating in the focus group interview were from several regions of the Republic of Korea and worked in LTC facilities with 68 to 109 older adult residents. None of the LTC facilities provided any systematic education programs for infection control, and infection control instruction was primarily performed by administrators or nursing managers if necessary. Education has been shown to improve compliance with infection control and reduce infection rates [16], and providing staff with education on COVID-19 is a critical factor for managing an outbreak [17]. The provision of in-service education on infection control reflects the level of preparation of LTC facilities to combat infectious disease, and the more prepared a facility is, the better it can combat the pandemic [18]. Therefore, facility administrators should develop and offer continuing in-service education programs to prevent and control infections. Moreover, it is necessary to understand the experiences of geriatric care workers during the COVID-19 pandemic and identify their educational needs for infection prevention.

The current results suggest that geriatric care workers primarily took charge of infection control tasks for emerging infectious diseases among all the staff working in the LTC facilities. Their job responsibilities related to infection control included checking whether patients were correctly wearing PPE, monitoring and reporting residents’ health conditions, cleaning and disinfecting their environment, ventilating, monitoring hand hygiene, and completing administrative documentation and paperwork.

However, geriatric care workers shared their difficulties and barriers related to infection control. They reported having increased workloads and difficulties with transferring patients to hospitals. Increased workloads related to COVID-19 are related to Limited staffing, which is the most commonly reported barrier to infection prevention and control [9]. Since higher staffing levels have been correlated with reduced COVID-19 outbreaks in LTC facilities [19], it is necessary to increase nurse staffing to prepare for emerging infectious diseases.

Geriatric care workers reported emotional distress during the implementation of pandemic-related visitor restrictions. When residents express feelings of loneliness and social isolation, geriatric care workers experience negative affectivity and job demands, which are related to negative job satisfaction and burnout [20,21]. Therefore, adequate psychological support to residents and staff was provided during infection outbreaks.

In addition, they had difficulties wearing PPE, following the existing infection control manual, and performing hand hygiene according to the manual. These reported experiences are consistent with previous research. Direct care providers in LTC facilities have limited access to PPE, hand hygiene, and cleaning products [9]. Thus, education on infection control and prevention should be provided to help resolve the difficulties geriatric care workers face [17]. Previous research found that geriatric care workers had high rates of infection control preventive behaviors when they received a relevant infection control education [22]. Sympathetic infection control education improves healthcare workers’ awareness of infection control and decreases incidences of infection [23]. Therefore, in this study, the educational needs of geriatric care workers were investigated to develop an infection control education.

In this study, the infection control educational needs of geriatric care workers were associated with the knowledge of a new infectious disease that was previously unknown. Frontline workers and leaders experienced knowledge gaps about infectious diseases and a lack of education about COVID-19 [17,24]. Furthermore, geriatric care workers reported a limited infection precaution and control education [9] and desired further education to alleviate their infection fears. Healthcare workers experience stress when they receive vague information about infections related to a novel virus [25]. An accurate and evidence-based education can reduce their fears and emotional burdens [25]. Therefore, providing an appropriate education about emerging infectious diseases should be included in the education programs for geriatric care workers in LTC facilities.

When asked about their preferences related to educational methods, geriatric care workers indicated that they preferred visual and practical teaching methods (e.g., video clips, repeated instruction, and clinical practice with demonstration). Since most infection control education content includes procedures or protocols addressing issues such as the correct use of PPE or administering a COVID-19 test, educational video clips, and demonstrational practices are effective methods of improving behavioral compliance with COVID-19 protocols [26]. Previous literature has suggested the need to develop infection control educational videos that can be distributed to LTC facilities during the COVID-19 pandemic [21]. In addition, education including repeated instruction leads to healthcare workers in LTC facilities practicing appropriate infection control procedures [27]. Therefore, repeatedly providing infection control education through video and clinical practice could maximize the effects of these educational programs for geriatric care workers in LTC facilities. Finally, health providers and institutions should advocate for changes in infection-related regulations and education systems to reflect the educational needs of geriatric care workers for infection control in LTC facilities.

This study has some limitations. The study participants were recruited from the southwest regions of the Republic of Korea using the snowball sampling method. Therefore, the findings should be interpreted and generalized cautiously to other populations. Another limitation may arise from the small sample size, potentially introducing the risk that data saturation was not achieved due to the limited number of participants. In addition, interviews were conducted using an online meeting platform (ZOOM) instead of face-to-face interview sessions due to restrictions on visiting the LTC facilities. Therefore, they may have been reluctant to express themselves as freely as they would have in an in-person meeting since the age group of participants was not familiar with using online platforms.

## 5. Conclusions

This qualitative and pilot study assessed the experiences of geriatric workers in LTC facilities attempting to control emerging infectious diseases, and identified their educational needs to assist with developing infection control educational programs related to emerging infectious diseases. To date, infection control education provided to geriatric care workers in LTC facilities has been primarily conducted by managers or department heads who lacked training in infection control. During the COVID-19 public health crisis, geriatric care workers experienced increased workloads and a variety of difficulties related to infection control processes, emotional distress, and transferring residents to hospitals.

Regarding their infection control education, geriatric care workers expressed their needs to gain broad in-depth knowledge regarding infectious diseases. If face-to-face education is difficult, the online education of infection control using e-learning videos needs to be developed, including practical skills. In addition, psychological support should be provided for geriatric care workers to relieve their fears and emotional distress regarding emerging infectious diseases.

## Figures and Tables

**Figure 1 healthcare-12-00301-f001:**
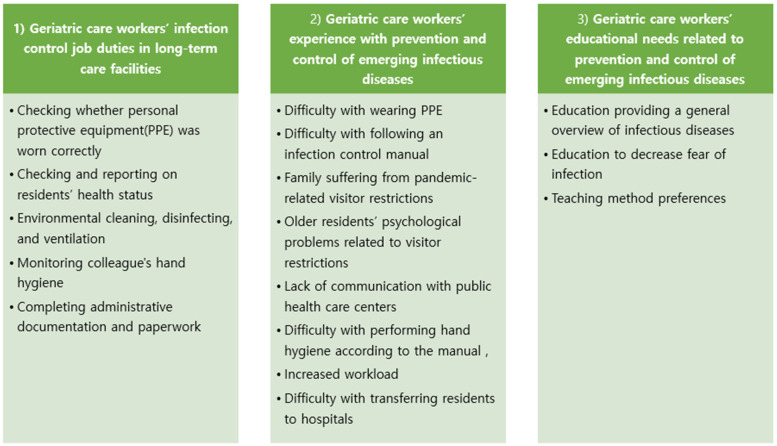
Summary of themes and sub-themes.

**Table 1 healthcare-12-00301-t001:** Focus group questions.

(1) Introduction question
- Before we start, can you introduce yourself?
(2) Transitional question
- Which infection control tasks are mainly performed by geriatric care workers in long-term care facilities currently?
(3) Main question
- What is your experience related to infection control in the workplace during the COVID-19 pandemic? - How has COVID-19 impacted your life and your patient’s care? - What kind of workplace infection education do you think is necessary for long-term care workers?
(4) Final question
- Is there anything else you want to talk about?

## Data Availability

All data used and analyzed during the current study are available from the corresponding author on request after a review of the purpose by the author.

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
