# Peer review of "Infection Control Experiences and Educational Needs of Geriatric Care Workers in Long-Term Care Facilities: A Pilot Study"

_healthcare, 2024, doi:10.3390/healthcare12030301_

Round 1

Reviewer 1 Report

Comments and Suggestions for Authors

Dear Authors

The article is based on an interesting topic for the scientific community and be used to improving the quality of care, worker’s health, and the continuity of research studies in such a crucial area and context.

However, the article has some weaknesses that need to be reviewed by the authors, and the following suggestions should be considered to improve quality.

Abstract: well-structured and balanced. The objective should be revised as it presents a different discourse to that mentioned at the end of the introduction section.

Introduction: It summarises the problem and the relevance of the topic, but it could be improved by elaborating the statement (p. 2, line 48) "in many countries....facilities".

Participants: It is unclear how potential participants were contacted.

Ethical Considerations: Was the study approved by the Institutional Review Board (p.29) with the information that participants "received a $15 reward in USD after completing the interview"? Why did the authors choose to give this reward? Did all the participants in the study benefit from it? What risks of response bias might be associated with this benefit?

Data Collection: It is unclear how long the interview lasted: 90 minutes or 2 hours? There is also lack of clarity about the questions included in the interview.

For how long and where were the interview recordings and transcript data kept ensuring data protection? How were they coded, per participant? This information is presented in the results section (e.g. Participant C) but it should be described here. They also mention "...selected important and meaningful..." what they mean by important and meaningful should be more explicit.

Results: Clearly presented and supported by figures to make them easier to read.

The discussion is simple, sustained and articulated with the objectives of the study and the results.

Limitations: The use of online recording may have compromised the answers - it is unclear why. Was this not considered when the participant's informed consent was obtained, and the reward for participating?

Conclusions: Should be more explicit when referring to systematic educational programmes, in addition to psychological support, considering the objectives of the study.

References: Relevant and up-to-date, but references 12 (2006), 15 (2011) and 22 (2010) should be revised and replaced with more recent evidence, considering the supported content.

Author Response

For research article

Response to Reviewer 1 Comments

1. Summary

2. Questions for General Evaluation

Reviewer’s Evaluation

Response and Revisions

Does the introduction provide sufficient background and include all relevant references?

Can e improved

Are all the cited references relevant to the research?

Can be improved

Is the research design appropriate?

Must be improved

Are the methods adequately described?

Must be improved

Are the results clearly presented?

Yes

Are the conclusions supported by the results?

Can be improved

3. Point-by-point response to Comments and Suggestions for Authors

Comments 1: Abstract: well-structured and balanced. The objective should be revised as it presents a different discourse to that mentioned at the end of the introduction section.

Response 1: Thank you for pointing this out. I agree with this comment. Therefore, I have revised paragraph 1 on page 1, line 14 (abstract, background) as below.

“In the post-COVID-19 condition, the importance of infection control education for geriatric care workers who care for the elderly who are vulnerable to emerging infectious diseases. This study was conducted to enhance insight into the experiences of geriatric care workers in managing novel infectious diseases(COVID-19) and to identify the newly required educational requirements necessary to effectively implement infectious disease control.”

Comments 2: Introduction: It summarises the problem and the relevance of the topic, but it could be improved by elaborating the statement (p. 2, line 48) "in many countries....facilities".

Response 2: Agree. I have revised the sentence to emphasize this point as follows– page 2, paragraph 2, and line 49.

Numerous nations have established and implemented infection prevention and control initiatives designed for personnel in LTC facilities.

Comments 3: Participants: It is unclear how potential participants were contacted.

Response: Thank you for pointing this out. I modified the sentence by adding the number of contacted people(page 2, paragraph 5, and line 85).

A researcher invited 10 geriatric care workers to participate in this study and answer related questions. All ten geriatric care workers in LTC facilities were recruited.”

Comments 4: Ethical Considerations: Was the study approved by the Institutional Review Board (p.29) with the information that participants "received a $15 reward in USD after completing the interview"? Why did the authors choose to give this reward? Did all the participants in the study benefit from it? What risks of response bias might be associated with this benefit?

Response 4: The $15 reward is intended as an expression of gratitude for your time devoted to the interview and is not associated with any benefits for the study. There is no inherent benefit or risk of response bias linked to this compensation. Moreover, the Institutional Review Board (IRB) has approved the study, which includes this reward.

I added one sentence about this on page 3, paragraph 1, line 102.

The reward is intended as an expression of gratitude for your time devoted to the interview and is not associated with any benefits for the study.”

Comments 5: Data Collection: It is unclear how long the interview lasted: 90 minutes or 2 hours?

Response 5: Agree. It was two hours. I have revised the sentence to emphasize this point as follows – page 3, paragraph 2, and line 111.

Focus group interviews were conducted twice, on September 15 and 16, 2021. The interviews lasted for two hours.”

Comments 6: There is also a lack of clarity about the questions included in the interview.

Response 6: Agree. I have added Table 1 (Focus group questions ) in page 3, paragraph 2, and line 120.

Comments 7: For how long and where were the interview recordings and transcript data kept ensuring data protection? How were they coded, per participant? This information is presented in the results section (e.g. Participant C) but it should be described here.

Response 7: I added more information about data protection in page 3, paragraph 1, line 104.

“Identification of participants was coded using pseudonyms, giving an initial letter per participant in alphabetical order. After the focus group interview, the data, including audio recordings and transcripts, were encrypted and secured using passwords and secure storage devices to prevent unauthorized access for three years, and after that, the data will be deleted and destroyed.”

Comments 8: They also mention "...selected important and meaningful..." what they mean by important and meaningful should be more explicit.

Response 8: Agree. I have revised the data analysis section to emphasize this point as follows – page 3, paragraph 3, and line 123.

“The analysis process was followed by the following main steps: 1) multiple sequential reading of the answers: the interview recordings were transcribed in a verbatim manner and thoroughly reviewed. 2) Extraction of the meaning codes: the statements underwent coding, categorization, and abstraction processes. 3) Categorization and abstraction: similar codes were clustered into the same sub-category. 4) Analysis and interpretation: the researchers collaborated to review one another’s opinions and agreed on the final categories derived.”

Comments 10: Results: Clearly presented and supported by figures to make them easier to read.

Response 10: Thank you.

Comments 11: The discussion is simple, sustained and articulated with the objectives of the study and the results.

Response 11: Thank you again.

Comments 12: Limitations: The use of online recording may have compromised the answers - it is unclear why. Was this not considered when the participant's informed consent was obtained, and the reward for participating?

Response12: Thank you for pointing this out. I agree with this comment. Therefore, I revised the sentence as follows – page 10, paragraph 4, and line 444.

In addition, interviews were conducted using an online meeting platform (ZOOM) instead of face-to-face interview sessions due to restrictions on visiting the LTC facilities. Therefore, they may have been reluctant to express themselves as freely as they would in an in-person meeting since the age group of participants was not familiar with using online.”

Comments 13: Conclusions: Should be more explicit when referring to systematic educational programmes, in addition to psychological support, considering the objectives of the study.

Response13: Agree. I modified the sentence as follows, – page 11, paragraph 1, and line 458.

Regarding their infection control education, geriatric care workers expressed their needs to gain broad, in-depth knowledge regarding infectious diseases. If face-to-face education is difficult, online education in infection control using e-learning videos needs to be developed, including practice skills. In addition, psychological support should be provided for geriatric care workers to relieve their fears and emotional distress regarding emerging infectious diseases.”

Comments 14: References: Relevant and up-to-date, but references 12 (2006), 15 (2011) and 22 (2010) should be revised and replaced with more recent evidence, considering the supported content.

Response 14:

Thank you for pointing this out. I changed them to the recent references– page 12, and lines 514, 525, and 549.

*To view the full manuscript, please see the attachment. 

Reviewer 2 Report

Comments and Suggestions for Authors

The article is interesting, and the topic has great relevance, but I suggest enriching the introduction by talking about all the problems related to healthcare-related infections. 

For example: liability and claims, multidrug resistance, risk management, vaccinations of health professionals

Also, the purpose of the analysis must be specified. 

In the text, the IRB number is blinded.

Have the study limitations?

To enrich the introduction I could suggest the following papers

Medico-Legal Aspects of Hospital-Acquired Infections: 5-Years of Judgements of the Civil Court of Rome

Treglia et al. Healthcare 2022

Protective Anti-HBs Antibodies and Response to a Booster Dose in Medical Students Vaccinated at Childhood

Coppeta et al. Vaccines 2023

Author Response

For research article

Response to Reviewer 2 Comments

1. Summary

2. Questions for General Evaluation

Reviewer’s Evaluation

Response and Revisions

Does the introduction provide sufficient background and include all relevant references?

Must be improved

Are all the cited references relevant to the research?

Must be improved

Is the research design appropriate?

Yes

Are the methods adequately described?

Can be improved

Are the results clearly presented?

Can be improved

Are the conclusions supported by the results?

Can be improved

3. Point-by-point response to Comments and Suggestions for Authors

Comments 1 : I suggest enriching the introduction by talking about all the problems related to healthcare-related infections. For example: liability and claims, multidrug resistance, risk management, vaccinations of health professionals

Response 1: Thank you for pointing this out.

For the next research topic, I will consider the research topic related to healthcare-related infections.

Comments 2: the purpose of the analysis must be specified. 

Response 2: Agree. I have revised the purposes as follows – page 2, paragraph 3, and line 73.  

This study aimed to conduct focus group interviews to understand the geriatric care workers' experiences while working to control new infectious diseases (i.e., COVID-19) and to identify their educational needs for developing systematic education programs regarding the newly required infectious diseases control.

Comments 3 : In the text, the IRB number is blinded.

Response 3: I blinded IRB numbers in the text on page 3, line 94.

Comments 4 : Have the study limitations?

Response 4: I wrote the study limitation on page 10, lines 440-448.

Comments 5 : To enrich the introduction I could suggest the following papers

Medico-Legal Aspects of Hospital-Acquired Infections: 5-Years of Judgements of the Civil Court of Rome Treglia et al. Healthcare 2022

Protective Anti-HBs Antibodies and Response to a Booster Dose in Medical Students Vaccinated at Childhood Coppeta et al. Vaccines 2023

Response 5: Thank you for giving me great articles.

I added the article in the introduction section on page 2, line 66.

No study has examined the educational needs of geriatric care workers in LTC facilities, although education was the most straightforward and effective measure to control infection [13, 14].”

*To view the full manuscript, please see the attachment

Reviewer 3 Report

Comments and Suggestions for Authors

The literature review in the introduction is sparse. The references are current but the content is inadequate. There are few references and some concepts are not clear in the introduction. The sample is small. 10 subjects seem too small a sample to be able to conclude from the results. The interview method is adequate but standardised assessments have not been used. It is advisable to complement the study with other more specific tests to reinforce the results. It is a study with little scientific basis and a small sample. 

Author Response

For research article

Response to Reviewer 3 Comments

1. Summary

2. Questions for General Evaluation

Reviewer’s Evaluation

Response and Revisions

Does the introduction provide sufficient background and include all relevant references?

Must be improved

Are all the cited references relevant to the research?

Must be improved

Is the research design appropriate?

Must be improved

Are the methods adequately described?

Must be improved

Are the results clearly presented?

Can be improved

Are the conclusions supported by the results?

Must be improved

3. Point-by-point response to Comments and Suggestions for Authors

Comments 1: The literature review in the introduction is sparse. The references are current but the content is inadequate. There are few references and some concepts are not clear in the introduction.

Response 1: Thank you for pointing this out. I agree with this comment.

Therefore, I rewrote the introduction by adding more references as follows, pages 1-2.

“The COVID-19 pandemic, which lasted for more than three years, was ended by the World Health Organization on May 5, 2023. Nevertheless, healthcare providers must be prepared to care for emerging infectious diseases like COVID-19. During the COVID-19 pandemic, it was found that older adults are vulnerable to emerging respiratory infectious diseases. The mortality rate in the ages of 80 years and above is very high (58.89%), with a fatality rate of 2.68%[1]. In particular, older adults in long-term care (LTC) facilities are at high risk of infectious disease outbreaks since most residents are medically fragile[2-4]. In the Republic of Korea, more than 65% of adults in LTC facilities were over age 80, and in the LTC facilities with confirmed cases of COVID-19, 65% reported clusters of confirmed cases [5].

According to policies related to COVID-19 infections in LTC facilities, case clusters were related to facilities having poor infection control systems and a lack of official guidelines or education programs addressing infectious disease prevention [5-6]. Moreover, older adults in LTC facilities are often at a heightened risk of infection due to difficulties in providing additional quarantine spaces, inadequate preventive measures, and insufficient resources to recover from COVID-19 [7]. Therefore, strategies to protect older adults living in LTC facilities from emerging infectious diseases such as COVID-19 infection are crucial, and infection control programs specializing in LTC facilities should be considered [6, 8].

Numerous nations have established and implemented infection prevention and control initiatives designed for personnel in LTC facilities. However, direct health care providers in LTC were more likely to perceive infection prevention and control as low priority rather than their responsibility. Furthermore, they have limited education about infection control and prevention [9]. In the Republic of Korea, geriatric care workers, certified by the Ministry of Health and Welfare, are responsible for providing care to infected patients. These workers have not received formal education from a professional medical institute but completed a short-term program lasting only 240 hours that includes theory and practice related to the care of frail older adults [10]. According to the training guidelines for geriatric care workers in LTC facilities, they are required to participate in continuing education and training once a year. However, these training programs do not include standardized educational guidelines for infection prevention and control, which do not exist [11].

Previous research has reported that having staff who are either underqualified or have inadequate infection control training represents a leading cause of increased COVID-19 spread [6]. Less than 3% of LTC facilities provided opportunities for healthcare workers to attend infection control courses compared to such opportunities that were available in over 90% of acute care hospitals [12]. No study has examined the educational needs of geriatric care workers in LTC facilities, although education was the most straightforward and effective measure to control infection [13, 14]. Given the significant impact of COVID-19 and other healthcare-associated infections in LTC, it is essential to understand the difficulties experienced by geriatric care workers related to infection control to better protect residents and geriatric care workers in LTC facilities. Furthermore, this study can be the baseline for developing infection control programs.

This study aimed to conduct focus group interviews to understand the geriatric care workers' experiences while working to control new infectious diseases (i.e., COVID-19) and to identify their educational needs for developing systematic education programs regarding the newly required infectious diseases control.”

Comments 2: The sample is small. 10 subjects seem too small a sample to be able to conclude from the results.

Response 2: Agree. I have, accordingly, searched the references related to the sample size of qualitative research and added the limitation of a small sample size.

According to the reference “Hennink M, Kaiser BN. Sample sizes for saturation in qualitative research: A systematic review of empirical tests. Soc Sci Med. 2022 Jan;292:114523. doi: 10.1016/j.socscimed.2021.114523.“, sample size recommendations is very wide: 5-60 interviews and 2-40 focus groups. In my study, there were two focus groups and five interviews of each group. I also added this as another limitation - page 10, line 442.

“Another limitation may arise from the small sample size, potentially introducing the risk that data saturation was not achieved due to the limited number of participants.”

Comments 3:

The interview method is adequate but standardised assessments have not been used. It is advisable to complement the study with other more specific tests to reinforce the results.

Response 3:

Thank you for pointing this out. I agree with this comment. Therefore, I modified the data analysis section to emphasize this point (page 3, line123) and added table 1 in the text.

“To analyze and organize the qualitative data, we used qualitative content analysis, as suggested by White and Marsh [13], which has been widely used in social sciences. The analysis process was followed by the following main steps: 1) multiple sequential reading of the answers: the interview recordings were transcribed in a verbatim manner and thoroughly reviewed. 2) Extraction of the meaning codes: the statements underwent coding, categorization, and abstraction processes. 3) Categorization and abstraction: similar codes were clustered into the same sub-category. 4) Analysis and interpretation: the researchers collaborated to review one another’s opinions and agreed on the final categories derived.

*To view the full manuscript, please see the attachment

Round 2

Reviewer 2 Report

Comments and Suggestions for Authors

I suggest to accept article in present form

Author Response

I appreciate your review of the paper.

Thanks to your feedback, the quality of the paper seems to have improved.

Thank you.

Reviewer 3 Report

Comments and Suggestions for Authors

The literature review in the introduction is sparse. The references are current but the content is inadequate. There are few references and some concepts are not clear in the introduction. The sample is small. 10 subjects seem too small a sample to be able to conclude from the results. The interview method is adequate but standardised assessments have not been used. It is advisable to complement the study with other more specific tests to reinforce the results. It is a study with little scientific basis and a small sample. I do not consider it adequate to have the content and scientific rigor to be published in this journal.

Author Response

Your feedback in round 2 mirrors that of round 1, where you remarked, 'The literature review in the introduction is sparse. The references are current but the content is inadequate. There are few references, and some concepts are not clear in the introduction. The sample is small. 10 subjects seem too small a sample to be able to conclude from the results. The interview method is adequate but standardized assessments have not been used. It is advisable to complement the study with other more specific tests to reinforce the results. It is a study with little scientific basis and a small sample.' I have previously addressed these concerns in my response. Thank you for your ongoing consideration.

Moreover, I deeply value your perspective expressed in round 2, 'I do not consider it adequate to have the content and scientific rigor to be published in this journal.' However, personally, I believe that this paper holds merit for presentation. Thank you sincerely for your comprehensive review of my work; your insights are truly appreciated.